# A New Hybrid Adaptive Self-Loading Filter and GRU-Net for Active Noise Control in a Right-Angle Bending Pipe of an Air Conditioner

**DOI:** 10.3390/s25206293

**Published:** 2025-10-10

**Authors:** Wenzhao Zhu, Zezheng Gu, Xiaoling Chen, Ping Xie, Lei Luo, Zonglong Bai

**Affiliations:** 1Key Laboratory of Intelligent Control and Neural Information Processing, Education Ministry of China, Yanshan University, Qinhuangdao 066004, China; wenzhaozhu@ysu.edu.cn (W.Z.);; 2School of Electrical Engineering, Yanshan University, Qinhuangdao 066004, China; 3Key Laboratory of Optoelectronic Technology and Systems, Education Ministry of China, Chongqing University, Chongqing 400044, China; 4Department of Electronic and Communication Engineering, North China Electric Power University, Baoding 071003, China

**Keywords:** active noise control, speech and air-conditioner noise, adaptive self-loading filter, hybrid deep recursive net

## Abstract

The air-conditioner noise in a rehabilitation room can seriously affect the mental state of patients. However, the existing single-layer active noise control (ANC) filters may fail to attenuate the complicated harmonic noise, and the deep recursive ANC method may fail to work in real time. To solve the problem, in a bending-pipe model, a new hybrid adaptive self-loading filtered-x least-mean-square (ASL-FxLMS) and convolutional neural network-gate recurrent unit (CNN-GRU) network is proposed. At first, based on the recursive GRU translation core, an improved CNN-GRU network with multi-head attention layers is proposed. Especially for complicated harmonic noises with more or fewer frequencies than harmonic models, the attenuation performance will be improved. In addition, its structure is optimized to decrease the computing load. In addition, an improved time-delay estimator is applied to improve the real-time ANC performance of CNN-GRU. Meanwhile, an adaptive self-loading FxLMS algorithm has been developed to deal with the uncertain components of complicated harmonic noise. Moreover, to achieve balance attenuation, robustness, and tracking performance, the ASL-FxLMS and CNN-GRU are connected by a convex combination structure. Furthermore, theoretical analysis and simulations are also conducted to show the effectiveness of the proposed method.

## 1. Introduction

People’s emotions are affected in environments with full-power air conditioners, especially patients in recovery [1]. An active noise control (ANC) system is an effective option to attenuate an air-conditioner noise [2]. Due to increasing noise attenuation needs in various settings, ANC technologies are continuously advancing, particularly concerning adaptive algorithms and network structures [3,4]. Presently, the ability to suppress complex noise has become a primary focus in enhancing ANC system performance [5].

Achieving a high attenuation level for certain complex noise remains an urgent ANC problem [6], especially for the Gaussian noise with multi-sinusoidal components. Due to the superior performance of Filtered-x Least-Mean-Square (FxLMS) methods, the Volterra expansions of FxLMS [7] are first used to reduce Gaussian noise with nonlinear sinusoidal components. Similarly, the Volterra Filtered-x Recursive Least Square (FxRLS) is applied to address weak non-linearity and modeling issues [8]. However, using too many parallel adaptive filters can cause interference in error signals, degrading the robustness of the ANC system. Therefore, an improved functional link neural network (FLANN) based on an artificial neural network (ANN) is proposed to attenuate each frequency independently [9]. Additionally, to enhance both attenuation and robustness [10], a feedback Volterra filter is employed to improve the accuracy of nonlinear path estimation [11].

However, real nonlinear noise always contains multiple frequencies, which are hard to predict with harmonic models. Therefore, the ANC system has introduced a neural network capable of handling random nonlinear problems [12]. Then, a Fuzzy neural network is used to attenuate the random single-frequency noise in ANC research [13]. Additionally, considering the correlations between previous and immediate inputs, a recurrent neural network is also employed in the ANC system [14]. However, the attenuation performance of these methods is lower than that of Volterra-based ANC methods.

Recently, a deep learning ANC (DANC) approach [15] has achieved better attenuation of certain noise than the traditional methods. It utilizes a convolutional layer and a Long Short-Term Memory (LSTM) network to classify and attenuate different kinds of noise. Significantly, the DANC can attenuate particular noises, e.g., babble and tire noises, which cannot be canceled well previously. Based on deep networks, feedback deep ANC structure [16], ANC with deep secondary path estimation [17], and ANC with deep weight vector estimation [18] are developed. However, considering the real-time performance, these deep network-based ANC methods need further improvements to decrease the computational load in one iteration of the ANC process. Although the LSTM network exhibits remarkable predictive capabilities in ANC application, the gate recurrent unit (GRU) performs a lower computational load [19]. Hence, the GRU-Net also has excellent potential in ANC applications. Additionally, real-world noise varies over time and includes known and unknown components [20]. The proportion of these components is not fixed, which can reduce the effectiveness of deep ANC methods. These methods require sufficient noise data to train their weight vectors for optimal performance. When a certain noise occurs briefly or has very low power, it becomes difficult to attenuate complex harmonic noises that contain multiple components [21]. In other words, the DANC method delivers exceptional attenuation mainly for the noise it has been trained on.

Actually, it can be seen that feedforward adaptive filters and deep neural networks have advantages and disadvantages. Hence, hybrid feedforward adaptive filters and deep neural networks ANC method are developed to balance the attenuation and generalization of ANC performance [22,23]. In particular, adaptive filters can handle complicated harmonic noise with uncertain components, while deep networks are suitable for periodic signals with certain components. Moreover, the convex combination is a good choice because the signal-to-noise ratio (SNR) of both certain and uncertain noise is difficult to estimate [24]. On the other hand, based on [18], a selective hybrid adaptive filter and deep network database algorithm are developed in [25], which provides another hybrid control structure to balance the attenuation and generalization performance. However, compared with the convex combined methods, the hybrid structure in [25] only has one part to control the weight vector. When certain and uncertain noise occur at nearly the same level, the method in [25] needs to select one controller from the adaptive filter and deep network database, which may fail to achieve good generalization performance.

As discussed above, to improve the attenuation and generalization performance simultaneously in different environments, a new hybrid adaptive self-loading FxLMS (ASL-FxLMS) and CNN-GRU structure is proposed for complicated harmonic noise control. Firstly, the CNN and GRU nets are serial-connected to achieve a good attenuation level and competitively low computing load for training certain complicated harmonic noise [15]. Then, the multi-head attention layer [26] is introduced to improve the tracking performance for complicated harmonic single frequencies. Moreover, an improved phase estimation is applied to estimate the time delay of the proposed CNN-GRU. On the other hand, inspired by the structure in [27] and the diagonal loading concept in [28], an ASL-FxLMS algorithm is developed for uncertain complicated harmonic noise. Furthermore, the ASL-FxLMS and CNN-GRU are connected by a convex combination structure [29]. This design allows both subsystems to produce outputs simultaneously. If one subsystem’s tracking performance and accuracy are insufficient, the other can compensate, leveraging the strengths of both subsystems. The contributions of each subsystem’s output are set based on amplitude, sustained amplitude, and SNR of the periodic and complex harmonic noise. This arrangement enables the combined system to balance attenuation, robustness, and generalization in tracking performance, potentially overcoming the synchronization output issues of the two subsystems [25]. Additionally, both certain and uncertain noises are used to evaluate the performance of the proposed method. To show a clear presentation of the proposed hybrid net, the novelty works are shown as follows:Inspired by the diagonal loading concept in [28], the adaptive self-loading FxLMS (ASL-FxLMS) is a newly proposed method.The CNN-GRU with attention mechanism is a new deep recursive neural network to be first applied in the ANC system, and a time-delay estimator is improved based on the reference [30].The framework combines ASL-FxLMS and CNN-GRU for complex noise environments, including both stable and complicated harmonic noise, has been developed.

Based on the references and previous discussions, the rest of the paper is organized as follows. At first, the hybrid structure of ASL-FxLMS and CNN-GRU-Net is illustrated in Section 2. Then, the CNN-GRU subsystem with multi-head attention and time-delay estimator is given in Section 3. In Section 4, the details and derivations of the ASL-FxLMS subsystem are illustrated. Moreover, in Section 5, numerical simulations and recording noise experiments are carried out to verify the superior performance of the proposed method. Finally, conclusions are given in Section 6.

## 2. Bending-Pipe Model of an Air Conditioner and the New Hybrid Active Noise Control Structure

### 2.1. Bending-Pipe Model of an Air Conditioner

In a rehabilitation room, when the air conditioner operates in variable frequency mode or at maximum power, the noise generated by the air conditioner is non-negligible. In addition, the brief diagram of the air-conditioner active noise control (ANC) system is shown in Figure 1. This noise contains complex harmonic frequencies and broadband time-varying noise, making it difficult for traditional noise reduction methods to address. Hence, based on [31], the Helmholtz equation is used to derive the propagation characteristics of sound waves in a curved pipe. Considering the mass conservation equation as(1)$$\frac{\partial \left(\rho_{0}+\tilde{\rho }\right)}{\partial t}+\nabla \left[\left(\rho_{0}+\tilde{\rho }\right)\nu \right]=0,$$
where $\tilde{\rho }$ is the perturbation of fluid density, $\rho_{0}$ is the constant fluid density, *t* is the time, and ν is the fluid velocity vector. Based on [31], the simplified Euler equations is given as(2)$$\rho_{0}\frac{\partial v}{\partial t}=-\nabla \tilde{p}.$$
where $\tilde{p}$ is the perturbation of sound pressure. Substituting Equation (Equation 2) into (Equation 1), we have(3)$$\nabla^{2}\tilde{p}-\left[\frac{\rho_{0}}{\tilde{\rho }\left(\frac{\partial^{2}\tilde{p}}{\partial t^{2}}\right)}\right]=0.$$
further reform the $\tilde{p}$ as the sum of the gradients of a scalar potential ϕ and a vector potential ψ as(4)$$\tilde{p}=\nabla \left(\phi +\psi \right).$$
transforming the ϕ and ψ into harmonic forms, we have(5)$$\phi =\phi_{r}e^{-j\omega t},$$
and(6)$$\psi =\psi_{r}e^{-j\omega t}.$$
re-substituting Equations (Equation 5) and (Equation 6) into Equation (Equation 4), the Helmholtz equation can be represented as(7)$$\nabla^{2}\left(\phi_{r}+\psi_{r}\right)+k^{2}\left(\phi_{r}+\psi_{r}\right)=0,$$
and(8)$$k=\frac{\omega }{c}.$$
where ω is the angular frequency, *k* is the wave number, and *c* is the speed of sound. Using MATLAB to solve Equation (Equation 7), the sound pressure distribution of the sound field in the curved pipe can be determined, and the propagation characteristics of the sound wave within the curved pipe can be analyzed. As in ref. [30], after determining the temperature and air density, the $\rho_{0}$ is calculated. Then, these values are entered into Equation (Equation 1), along with the propagation length *r*, to obtain Equation (Equation 7) and find the solution. Repeat these steps until the secondary path model is established. Input white noise to train the secondary path model and obtain the secondary path parameters for the bent pipe model.

### 2.2. Hybrid of ASL-FxLMS and CNN-GRU

As mentioned in ref. [15], the deep CNN-LSTM network has great attenuation potential for certain noise that can be recorded. However, uncertain noise is a kind of interference for the deep CNN-LSTM, which may deteriorate its robustness. To solve the problem, a hybrid structure with different types of adaptive networks is proposed as shown in Figure 2. The main purpose of the proposed method is to integrate the complex memory capabilities of deep learning networks with the flexibility of adaptive filters to enhance real-time ANC systems, thereby overcoming issues with mixture noise of stable noise and unstable noise. That is, the real-time single-layer adaptive filter and deep learning network are connected through a convex combination structure. The new system will have the advantages of two subsystems. Moreover, the proposed method integrates embedded systems and computer systems to adapt to practical applications. As shown in Figure 2, the first part is the ASL-FxLMS (Adaptive Self-Loading Filtered-x Least-Mean-Square) subsystem in a blue dashed line. The second part is the CNN-GRU (Convolutional Neural Network–Gated Recurrent Unit) subsystem in a red dashed line. The two subsystems are illustrated in detail in Section 3 and Section 4. In this hybrid structure, ASL-FxLMS is designed to attenuate uncertain noise with strong harmonic characteristics. The CNN-GRU is applied to attenuate certain noise in the time domain with periodical characteristics. Specifically, the hybrid convex combination structure is applied to balance robustness and attenuation performance, especially in a time-varying environment.

For the ASL-FxLMS in the Figure 2, the output $y_{A}\left(n\right)$ of a single-layer adaptive weight vector is represented as $\Theta \left(n\right)$. The updating equation is shown as(9)$$\Theta \left(n+1\right)=\Theta \left(n\right)+\mu_{A}x_{f}\left(n\right)e_{A}\left(n\right),$$
where $\mu_{A}$ is the step size, $e_{A}\left(n\right)$ is the error signal of ASL-FxLMS subsystem, given by(10)$$e_{A}\left(n\right)=\hat{d}\left(n\right)-{\hat{y}}_{\mathrm{Af}}\left(n\right),$$
where ${\hat{y}}_{\mathrm{Af}}\left(n\right)$ is the filtered signal which is the result of $y_{\mathrm{Af}}\left(n\right)$ filtered by the estimated secondary path $\hat{S}\left(z\right)$, $\hat{d}\left(n\right)$ is the estimated reference signal.

As for the CNN-GRU-Net in the red-line part in the Figure 2, the $y_{\mathrm{CG}}$ is the output of the CNN-GRU-Net. Moreover, the output after time-delay compensation is ${\bar{y}}_{\mathrm{CG}}\left(n\right)$. Furthermore, after the ${\bar{y}}_{\mathrm{CG}}\left(n\right)$ is filtered by the estimated secondary path $\hat{S}\left(z\right)$, the filtered signal ${\hat{y}}_{\mathrm{CGf}}\left(n\right)$ is also calculated. Hence, the error $e_{\mathrm{CG}}$ of CNN-GRU for fine-tune process of CNN-GRU is represented as(11)$$e_{\mathrm{CG}}\left(n\right)=\hat{d}\left(n\right)-{\hat{y}}_{\mathrm{CGf}}\left(n\right),$$
where the fine-tune uses an adaptive strategy, the additional layer is set before the final full-connection layers.

Then, the output $y_{\mathrm{Hy}}\left(n\right)$ of the hybrid structure is represented as(12)$$y_{\mathrm{Hy}}\left(n\right)=\lambda \left(n\right)y_{A}\left(n\right)+\left[1-\lambda \left(n\right)\right]{\bar{y}}_{\mathrm{CG}}\left(n\right).$$
where the $\lambda \left(n\right)$ is shown as(13)$$\lambda \left(n\right)=\frac{1}{1+e^{-\sigma \left(n\right)}},$$
and $\sigma \left(n\right)$ is updated as the same way in ref. [24] as(14)$$\begin{array}{c} \sigma \left(n+1\right)=\sigma \left(n\right)\\ +\mu_{\sigma }\frac{2e\left(n\right)}{{\left[1+e^{2}\left(n\right)\right]}^{2}}\left[{\bar{y}}_{\mathrm{CG}}\left(n\right)-y_{A}\left(n\right)\right]\lambda \left(n\right)\left[1-\lambda \left(n\right)\right], \end{array}$$
where $\mu_{\sigma }$ is the step size.

Subsequently, as shown in the green-line part in Figure 2, the output $y_{\mathrm{Hy}}\left(n\right)$ of the secondary loud-speaker is sent into the real acoustic path. After filtering by the real secondary acoustic path $s\left(z\right)$, the filtered signal $y_{\mathrm{Hyf}}\left(n\right)$ is subtracted by the real reference signal $d\left(n\right)$ at the error sensor. Hence, the real error $e\left(n\right)$ is received by the error sensor.

Meanwhile, to estimate the reference signal and error signal for both ASL-FxLMS and CNN-GRU-Net, the output $y_{\mathrm{Hy}}\left(n\right)$ is filtered by the estimated secondary path $\hat{S}\left(z\right)$. In addition, the ${\hat{y}}_{\mathrm{Hyf}}\left(n\right)$ is the filtered signal. Hence, the estimated reference signal $\hat{d}\left(n\right)$ at the error sensor is represented as(15)$$\hat{d}\left(n\right)=e\left(n\right)+{\hat{y}}_{\mathrm{Hyf}}\left(n\right).$$

Further details of the CNN-GRU subsystem and ASL-FxLMS subsystem are given in the following sections. The main notations of the proposed hybrid ASL-FxLMS and CNN-GRU are shown in Table 1.

## 3. A CNN-GRU Network with Multi-Head Attention Mechanism

Benefiting from the powerful nonlinear mapping capabilities of neural networks [32,33,34], it has been extensively used in ANC systems to enhance noise reduction performance [35,36]. Nevertheless, real-world challenges such as real-time constraints, acoustic path variations, and shifts like background noise have imposed limitations on its effectiveness [37]. As a result, there remains a critical need to strike a balance between the noise reduction capability and real-time performance within the ANC framework using deep networks. In the proposed work, nonlinear noise means [2], where, compared with the harmonic frequency model, some frequencies are missing, resulting in noise that cannot be accurately modeled by a linear model for all frequencies.

### 3.1. Pre-Train: The CNN-GRU-Net with Multi-Head Attention

To solve the issue of denoising irregular harmonic frequency noise in practical applications, an improved CNN-GRU was proposed, as shown in Figure 3, to handle a type of noise that contains multiple single-frequency noises and has unclear harmonic frequency components [2]. In Figure 3, the complex structure of the CNN-GRU network is depicted through a series of carefully coordinated steps.

(I) Input data: Divide the time series into multiple groups. Apply the Fourier transform to each group. Extract the necessary frequency information from the noise.

(II) Data compression: Use a CNN network to classify different frequency groups, and apply network parameter settings to filter out infrequent frequency information.

(III) Encoding-GRU-decoding: The main structure of the GRU network encodes essential frequency group information and makes predictions. Compared to the LSTM in [15], it has a simpler design and lower computational requirements. After prediction, the result is decoded to extract the frequency group information that needs to be tracked.

(IV) Single-frequency noise in real-world applications can result from factors like mechanical vibration, material rigidity, and different populations. As a result, the single-frequency component in actual scenarios will be much higher than in theoretical analysis and simulations. Based on the multi-head attention mechanism in [26], the proposed CNN-GRU will reduce the number of frequencies needing attention and enhance the noise reduction capacity of the main frequency group. More specifically, the detailed processes of the multi-head attention layer are shown in Figure 4, where the *K*, *Q*, and *V* represent Key, Query, and Value, respectively. In the step 1, the similarity of *K* and *Q* is calculated as(16)Sum=Q·K.
then, the vector Sum is normalized by the SoftMax function as(17)$$W=\mathrm{Softmax}\left(\mathrm{Sum}\right),$$
where(18)$$\mathrm{Sum}=\left[Sum1,Sum2,\cdots ,Sumi,\cdots ,SumI\right],$$
and(19)$$W=\left[w_{1},w_{2},\cdots ,w_{i},\cdots ,w_{I}\right],$$
and *I* represents the total number of frequencies. While the output of the multi-head attention layer can be represented as(20)AttentionOutput=W·V.

(V) Decompressing data and reconstructing time series: In the final stage, decompression and output data processing help convert the signal from the abstract frequency domain back to the concrete time domain. Here, fine-tuning methods are introduced in the final fully connected layer to improve the system’s responsiveness and adaptability in real-time scenarios.

### 3.2. Real-Time Control: Improved Time-Delay Estimator

The proposed CNN-GRU needs noise data and enough training time to achieve noise tracking. However, a phase error still exists [15], which deteriorates the ANC robustness. More exactly, in the real acoustic environment, the time delay will change with time. Hence, an improved time-delay estimator is applied based on the time-delay estimator in ref. [30]. And the CNN-GRU network is pre-trained, which means it may fail to track the complicated harmonic delay. Thus, based on this characteristic, an auxiliary pre-Fourier analyzer and an Error-Fourier analyzer are combined with the phase tracking filter as in Figure 5. The three parts of the additional phase tracking filter are illustrated as follows.

#### 3.2.1. Pre-Fourier Analyzer

The Pre-Fourier analyzer is shown in green parts in Figure 5. It is used to extract the time-domain signal with the strongest single-frequency noise over a period of time. At first, the $x\left(n\right)$, which is from the noise source, is input as a reference signal for the Pre-Fourier analyzer. Simultaneously, the $x\left(n\right)$ is put into a time sequence, which is named as Buffer in the Figure 5. After 256 point FFT (Fast Fourier Transform), the single-frequency noise with max-power is signed to provide reference $\omega_{i}$. More exactly, the reference ω will change with time, and several single-frequency noises are possible to be set as the reference frequency. Thus, the reference $\omega_{i}$ is used rather than ω to show that the reference frequency will change with time. Based on the above, the error $e_{\mathrm{PFa}}\left(n\right)$ of Pre-Fourier analyzer is given as(21)$$e_{\mathrm{PFa}}\left(n\right)=x\left(n\right)-\left[{\hat{a}}_{c,i}\left(n\right)x_{ai}\left(n\right)+{\hat{b}}_{c,i}\left(n\right)x_{bi}\left(n\right)\right],$$
where $x_{ai}\left(n\right)$ represents for $\cos\left(\omega_{i}n\right)$ and $x_{bi}\left(n\right)$ represents for $\sin\left(\omega_{i}n\right)$. Moreover, using the LMS (Least-Mean-Square) algorithm to update the weight vector ${\hat{a}}_{c,i}$ and ${\hat{b}}_{c,i}$ as(22)$${\hat{a}}_{c,i}\left(n+1\right)={\hat{a}}_{c,i}\left(n\right)+\mu_{\mathrm{PFa}}x_{ai}\left(n\right)e_{\mathrm{PFa}}\left(n\right)$$
and(23)$${\hat{b}}_{c,i}\left(n+1\right)={\hat{b}}_{c,i}\left(n\right)+\mu_{\mathrm{PFa}}x_{bi}\left(n\right)e_{\mathrm{PFa}}\left(n\right),$$
respectively. $\mu_{\mathrm{PFa}}$ is the step size. Furthermore, the output $y_{\mathrm{PFa}}\left(n\right)$ of Pre-Fourier analyzer is given as(24)$$y_{\mathrm{PFa}}\left(n\right)={\hat{a}}_{c,i}\left(n\right)x_{ai}\left(n\right)+{\hat{b}}_{c,i}\left(n\right)x_{bi}\left(n\right).$$

#### 3.2.2. Error-Fourier Analyzer

In the Error-Fourier analyzer process, which is shown as the red part in Figure 5, the reference $\omega_{i}$ is provided by the Pre-Fourier analyzer process. Moreover, the calculation formulas for the Error-Fourier and Pre-Fourier analyzer are similar in principle. Hence, the output and updating equations of the Error-Fourier analyzer process are as follows. At first, the error of $e_{\mathrm{EFa}}\left(n\right)$ of Error-Fourier analyzer is given as(25)$$e_{\mathrm{EFa}}\left(n\right)=e_{\mathrm{CG}}\left(n\right)-\left[{\hat{a}}_{e,i}\left(n\right)x_{ai}\left(n\right)+{\hat{b}}_{e,i}\left(n\right)x_{bi}\left(n\right)\right].$$
and the updating equations is given as(26)$${\hat{a}}_{e,i}\left(n+1\right)={\hat{a}}_{e,i}\left(n\right)+\mu_{\mathrm{EFa}}x_{ai}\left(n\right)e_{\mathrm{EFa}}\left(n\right),$$(27)$${\hat{b}}_{e,i}\left(n+1\right)={\hat{b}}_{e,i}\left(n\right)+\mu_{\mathrm{EFa}}x_{bi}\left(n\right)e_{\mathrm{EFa}}\left(n\right),$$
where $\mu_{\mathrm{EFa}}$ is the step size in Error-Fourier analyzer. In addition, the output $y_{\mathrm{EFa}}\left(n\right)$ of Error-Fourier analyzer is given as(28)$$y_{\mathrm{EFa}}\left(n\right)={\hat{a}}_{e,i}\left(n\right)x_{ai}\left(n\right)+{\hat{b}}_{e,i}\left(n\right)x_{bi}\left(n\right).$$

#### 3.2.3. Phase Tracking Filter

Although the deep network can learn more noise characteristics than single adaptive filters using the reference and error signal, it takes a longer time to converge, which leads to phase error. To solve this problem, a repetitive control algorithm is introduced to achieve optimal phase tracking as the blue part in Figure 5. The output $y_{\mathrm{PFa}}\left(n\right)$ of the Pre-Fourier analyzer is input into a buffer and changes as $y_{\mathrm{PFa}}\left(n\right)$. Similarly, the $y_{\mathrm{EFa}}\left(n\right)$ is input into a buffer and changes as $y_{\mathrm{EFa}}\left(n\right)$. Then the error $e_{\mathrm{Ptf}}\left(n\right)$ of phase tracking filter is given as(29)$$e_{\mathrm{Ptf}}\left(n\right)=y_{\mathrm{PFa}}\left(n\right)-y_{\mathrm{EFa}}\left(n\right).$$

To estimate the time delay of the secondary path, an $N_{m}$ order FIR (Finite Impulse Response) filter is applied. The output is given as(30)$$y_{\mathrm{EFa}}\left(n\right)=q\left(y_{\mathrm{EFa}}\left(n-p\right)-\sum_{m=1}^{N_{m}}l\left(n\right)y_{\mathrm{IL}}\left(n-m+1\right)\right),$$
where *q* is the amplitude compensation coefficient and *p* is the time-delay compensation factor, $l\left(n\right)$ is an FIR filter to adjust the time delay, given by(31)$$l\left(n\right)=e^{-j2\pi fn},$$
where f represents the frequency vector. $y_{\mathrm{IL}}\left(n\right)$ is defined as(32)$$y_{\mathrm{IL}}\left(n\right)=\sum_{m=1}^{N_{m}}e_{\mathrm{Ptf}}\left(n\right)\hat{s}\left(n-N_{m}+1\right),$$
where $\hat{s}\left(n\right)$ represents the *n*-th coefficient of the estimated secondary path, and $N_{m}$ represents half-length of the estimated secondary path. Moreover, the $y_{\mathrm{IL}}\left(n\right)$ is the vector form of $y_{\mathrm{IL}}\left(n\right)$.

### 3.3. Convergence Condition Analysis of CNN-GRU Subsystem in ANC Process

Although it is impossible to analyze the steady-state and convergence condition of CNN-GRU step by step in the pre-training process, the convergence condition of the ANC process can be analyzed. As shown in Figure 2, the CNN-GRU has an adaptive fine-tune filter updated by the error signal $e_{\mathrm{CG}}\left(n\right)$. The detailed connection of CNN-GRU and fine-tune filter is shown in Figure 6. As the same function is illustrated in [15], the fine-tune filter is set to modify the output of the deep network. In the paper, the 256-length reference signal is an input block. The CNN-GRU network is trained offline with 80 types of noise. Moreover, a 256-length adaptive filter is introduced as the fine-tune weight vector to adjust the output of CNN-GRU to fit the acoustic environment changes.

In the ANC process, the weight vector $\Theta_{\mathrm{CG}}\left(n\right)$ of CNN-GRU is fixed. An updating equation of the 256 length fine-tune adaptive filter is given as(33)$$\Theta_{\mathrm{Ft}}\left(n+1\right)=\Theta_{\mathrm{Ft}}\left(n\right)+\mu_{\mathrm{Ft}}e_{\mathrm{CG}}\left(n\right)x\left(n\right).$$
where $\mu_{\mathrm{Ft}}$ is the step size of fine-tune filter, $e_{\mathrm{CG}}\left(n\right)$ is the error signal, $\Theta_{\mathrm{Ft}}\left(n\right)$ is the weight vector of fine-tune filter, and $x\left(n\right)$ represent the vector of $x\left(n\right)$. Considering the time-delay estimator as shown in the yellow block in Figure 2, the output of GNN-GRU is(34)$$\hat{d}\left(n\right)\approx \left\{\Theta_{\mathrm{CG}}^{T}+\Theta_{\mathrm{Ft}}^{T}\left(n\right)\right\}x_{f}\left(n\right).$$
and define the error weight vector ${\tilde{\Theta }}_{\mathrm{Ft}}\left(n\right)$ of fine-tune filter as(35)$${\tilde{\Theta }}_{\mathrm{Ft}}\left(n\right)=\Theta_{\mathrm{CG}}+\Theta_{\mathrm{Ft}\_\mathrm{Opt}}-\left[\Theta_{\mathrm{CG}}+\Theta_{\mathrm{Ft}}\left(n\right)\right].$$
thus, the error signal $e_{\mathrm{CG}}\left(n\right)$ is shown as(36)$$\begin{array}{c} e_{\mathrm{CG}}\left(n\right)=\hat{d}\left(n\right)-{\bar{y}}_{\mathrm{CG}}\left(n\right)=\hat{d}\left(n\right)-y_{\mathrm{CG}}\left(n\right)e^{-2j\pi fT}\approx {\tilde{\Theta }}_{\mathrm{Ft}}^{T}\left(n\right)x\left(n\right) \end{array}.$$
where *f* is the frequency, *T* is the time delay. To obtain further steady-state equations [27], taking the Euclidean norm on both sides of Equation (Equation 33), we have(37)$$\begin{array}{c} E\left[{\tilde{\Theta }}_{\mathrm{Ft}}^{2}\left(n+1\right)\right]=E\left[{\tilde{\Theta }}_{\mathrm{Ft}}^{2}\left(n\right)\right]+\mu_{\mathrm{Ft}}^{2}E\left[e_{\mathrm{CG}}^{2}\left(n\right)x^{2}\left(n\right)\right]+2\mu_{\mathrm{Ft}}E\left[{\tilde{\Theta }}_{\mathrm{Ft}}^{T}\left(n\right)x\left(n\right)e_{\mathrm{CG}}\left(n\right)\right] \end{array}.$$

To keep robust, limited by the following equations,(38)$$E\left[{\tilde{\Theta }}_{\mathrm{Ft}}^{2}\left(n+1\right)\right]\leq E\left[{\tilde{\Theta }}_{\mathrm{Ft}}^{2}\left(n\right)\right].$$
further considering the real noise power is not normalized, hence the step size $\mu_{\mathrm{Ft}}$ should be limited by(39)$$0<\mu_{\mathrm{Ft}}<\frac{2}{e^{-4j\pi fT}\lambda_{x}^{2}},$$
where $\lambda_{x}^{2}$ is the power of $x\left(n\right)$. In addition, further limited by(40)$$e^{-4j\pi fT}\leq 1.$$Equation (Equation 39) can be rewritten as(41)$$0<\mu_{\mathrm{Ft}}<\min\left(\frac{2}{e^{-4j\pi fT}\lambda_{x}^{2}}\right)=\frac{2}{\lambda_{x}^{2}}.$$

## 4. Adaptive Self-Loading FxLMS Subsystem

To deal with air-conditioner noise with power changes, a clear method is to set another parallel adaptive filter [38]. However, the CNN-GRU has superior attenuation performance for harmonic and periodical noise, which may lead to lower energy of stable harmonic signals in error signals. This will influence the robustness of the parallel adaptive filter. Hence, the parallel filter should have good robustness even if there is no periodic noise in the error signal. Based on all mentioned, as shown in Figure 7, an adaptive self-loading filtered-x least-mean-square (ASL-FxLMS) algorithm is proposed in this section for another parallel adaptive filter.

### 4.1. The New Adaptive Self-Loading FxLMS with Online Secondary Path Estimation Strategy

Considering the parallel CNN-GRU mentioned in Section 3, sometimes there will be no periodic components in the error signal, which will deteriorate the tracking performance of the FxLMS. As mentioned above, based on the online estimation structure in [27] and the diagonal loading method in [28], a self-loading filter is introduced to improve the robustness of the ANC system.

As shown in the dark green part in Figure 7, the proposed adaptive self-loading filter uses the online estimation structure to inject a known sinusoidal wave into the reference signal $x\left(n\right)$ as the input for the weight vector $\Theta \left(n\right)$. In this way, the self-loading filter stabilizes the tracking performance of the filter by forcibly inputting maximum sinusoidal periodical signals. In the initiate state, the reference signal $x\left(n\right)$ and output $y_{\mathrm{Slf}}\left(n\right)$ of the adaptive self-loading filter is added as input time-series in the reference input array $x\left(n\right)$ as(42)$$\begin{array}{c} x\left(n\right)=\left[x\left(n\right)+y_{\mathrm{Slf}}\left(n\right),x\left(n-1\right)+y_{\mathrm{Slf}}\left(n-1\right),\cdots ,x\left(n-m_{x}+1\right)+y_{\mathrm{Slf}}\left(n-m_{x}+1\right)\right], \end{array}$$
where $m_{x}$ is the length of input array $x\left(n\right)$. Then, filtered by the single-layer adaptive filter $\Theta \left(n\right)$, the output $y_{A}\left(n\right)$ of $\Theta \left(n\right)$ is given as(43)$$y_{A}\left(n\right)=\Theta^{T}\left(n\right)x\left(n\right).$$
moreover, filtered by the secondary path $s\left(n\right)$, the ${\hat{y}}_{\mathrm{Af}}\left(n\right)$ is represented as(44)$${\hat{y}}_{\mathrm{Af}}\left(n\right)=\sum_{m=1}^{m_{S}}\left[y_{A}\left(n\right)-y_{\mathrm{Slf}}\left(n\right)-N_{\mathrm{OS}}\left(n\right)\right]s\left(n-m\right),$$
where(45)$$N_{\mathrm{OS}}\left(n\right)=N\left(n\right)\left|e_{A}\left(n-1\right)\right|,$$
and $s\left(n\right)$ is the coefficient of secondary path FIR filter. In addition, $m_{S}$ is the length of the secondary path FIR filter. Then, the error $e_{A}$ of single-layer filter is(46)$$e_{A}\left(n\right)=\hat{d}\left(n\right)-{\hat{y}}_{\mathrm{Af}}\left(n\right).$$

Based on the above equations and steady-state assumption in [27], using the FxLMS method to update the weight vectors, the updating equations of the noise control filter $\Theta \left(n\right)$ is given as(47)$$\Theta \left(n+1\right)=\Theta \left(n\right)+\mu_{A}e_{A}\left(n\right)x_{f}\left(n\right),$$
where $\mu_{A}$ is the step size of the noise control filter.

Moreover, for the self-loading filter, the output $y_{\mathrm{Slf}}\left(n\right)$ can be represented as(48)$$y_{\mathrm{Slf}}\left(n\right)={\hat{a}}_{s,i}\left(n\right)x_{ai}\left(n\right)+{\hat{b}}_{s,i}\left(n\right)x_{bi}\left(n\right),$$
and hence the error $e_{\mathrm{Slf}}\left(n\right)$ is represented as(49)$$e_{\mathrm{Slf}}\left(n\right)=x\left(n\right)-y_{\mathrm{Slf}}\left(n\right).$$
meanwhile, the updating equation of the self-loading filter is represented as(50)$${\hat{a}}_{s,i}\left(n+1\right)={\hat{a}}_{s,i}\left(n\right)+\mu_{\mathrm{Slf}}x_{ai}\left(n\right)e_{\mathrm{Slf}}\left(n\right)$$
and(51)$${\hat{b}}_{s,i}\left(n+1\right)={\hat{b}}_{s,i}\left(n\right)+\mu_{\mathrm{Slf}}x_{bi}\left(n\right)e_{\mathrm{Slf}}\left(n\right).$$

Furthermore, for the filter of online secondary path estimation, the updating equations of the weight vector are represented as(52)$$\hat{S}\left(n+1\right)=\hat{S}\left(n\right)+\mu_{\mathrm{OS}}e_{\mathrm{OS}}\left(n\right)N_{\mathrm{OS}}\left(n\right),$$
where(53)$$e_{\mathrm{OS}}\left(n\right)=e_{A}\left(n\right)-\sum_{m=1}^{m_{S}}\left[N_{\mathrm{OS}}\left(n\right)\hat{s}\left(n-m\right)\right],$$
and $\hat{s}\left(n\right)$ is the coefficient of weight vector $\hat{S}\left(n\right)$.

### 4.2. Convergence Condition Analysis

To ensure the robustness of the ASL-FxLMS algorithm, a steady-state analysis must be performed that provides proper step-size ranges. Assuming that the $\Theta_{\mathrm{Opt}}$ is the optimal weight vector in steady states. In addition, the approximation estimation of $\hat{d}\left(n\right)$ is given as follows:(54)$$\hat{d}\left(n\right)\approx \Theta_{\mathrm{Opt}}^{T}x_{f}\left(n\right).$$
then the error weight vector $\tilde{\Theta }\left(n\right)$ between optimal weight vector $\Theta_{\mathrm{Opt}}$ and real-time weight vector $\Theta \left(n\right)$ is represented as(55)$$\tilde{\Theta }\left(n\right)=\Theta_{\mathrm{Opt}}-\Theta \left(n\right).$$
and as shown in Equation (Equation 46), the error signal $e_{A}\left(n\right)$ of the noise control filter of ASL-FxLMS can also be estimated by(56)$$e_{A}\left(n\right)=\hat{d}\left(n\right)-{\hat{y}}_{\mathrm{Af}}\left(n\right)\approx {\tilde{\Theta }}^{T}\left(n\right)x_{f}\left(n\right).$$
by subtracting the optimal weight vector $\Theta_{\mathrm{Opt}}$ on both side of Equation (Equation 47), we obtain(57)$$\tilde{\Theta }\left(n+1\right)=\tilde{\Theta }\left(n\right)+\mu_{A}e_{A}\left(n\right)x_{f}\left(n\right).$$

To obtain further steady-state equations, taking the Euclidean norm on both sides of Equation (Equation 57), we have(58)$$\begin{array}{c} {\tilde{\Theta }}^{2}\left(n+1\right)={\tilde{\Theta }}^{2}\left(n\right)+\mu_{A}^{2}e_{A}^{2}\left(n\right)x_{f}^{2}\left(n\right)+2\mu_{A}\tilde{\Theta }\left(n\right)e_{A}\left(n\right)x_{f}\left(n\right). \end{array}$$

Assuming that the weight vector changes slowly, meanwhile, taking expectations on both sides of Equation (Equation 58), we obtain(59)$$\begin{array}{c} E\left[{\tilde{\Theta }}^{2}\left(n+1\right)\right]=E\left[{\tilde{\Theta }}^{2}\left(n\right)\right]+\mu_{A}^{2}E\left[e_{A}^{2}\left(n\right)x_{f}^{2}\left(n\right)\right]+2\mu_{A}E\left[{\tilde{\Theta }}^{T}\left(n\right)x_{f}\left(n\right)e_{A}\left(n\right)\right]. \end{array}$$

To keep robust, limited by the following equations,(60)$$E\left[{\tilde{\Theta }}^{2}\left(n+1\right)\right]\leq E\left[{\tilde{\Theta }}^{2}\left(n\right)\right].$$
the step size $\mu_{A}$ should be limited by(61)$$0<\mu_{A}\leq 2.$$

Considering that the real noise power is not normalized. Hence Equation (Equation 61) should be rewrite as(62)$$0<\mu_{A}\leq \min\left(2,\frac{2}{{\lambda_{x}}^{2}}\right),$$
where ${\lambda_{x}}^{2}$ is the power of input reference signal $x\left(n\right)$.

Moreover, by similar calculations, the range of step size $\mu_{\mathrm{Slf}}$ is represented as(63)$$0<\mu_{\mathrm{Slf}}\leq \frac{2}{{\lambda_{x,i}}^{2}}.$$
where the ${\lambda_{x,i}}^{2}$ is the power of the maximum power frequency noise. Furthermore, the range of step size $\mu_{\mathrm{OS}}$ is represented as(64)$$0<\mu_{\mathrm{OS}}\leq \frac{2}{\lambda_{N}^{2}\lambda_{x}^{2}},$$$\lambda_{N}^{2}$ is the power of injecting Gaussian white noise.

## 5. Simulations

Several simulations are conducted in this section to verify the attenuation performance for the complicated harmonic noise. In addition, standard FxLMS [2], FLANN [9], and DANC [15] algorithms are introduced as comparisons of the proposed hybrid ASL-FxLMS and CNN-GRU. Moreover, the step size of FxLMS, FLANN, and ASL-FxLMS is set as 0.002. More exactly, the structure of CNN-GRU with multi-head attention is shown in Figure 8. In the same way as the parameters shown in [15], the input size and output size are defined as FeatureMaps×TimeSteps×FrequencyNumbers, the hyperparameter is defined as KernelSize×Srides×OutputChannel. In addition, the parameters of the proposed CNN-GRU are shown in Table 2. The computational cost of each mentioned algorithm in one step is given in Table 3. The main operation of the proposed hybrid CNN-GRU and ASL-FxLMS is shown in Table 4.

The noise-92 datasets are used to train the CNN-GRU model, like the operation in [15]. The following simulations involve a changing noise environment and the changing acoustic paths as shown in Figure 9 and Figure 10. The hybrid embedded and computer ANC system framework is tested here as shown in Figure 11. More precisely, the computation of a convex combination is set in a Field Programmable Gate Array (FPGA). Meanwhile, the FPGA updates the adaptive weight vectors. The computing of CNN-GRU is set in the CPU and GPU of a computer. Moreover, the PCI-E (Peripheral Component Interconnect Express) connects the embedded and computer systems. It should also be noted that the definition of the SNR (Signal-to-Noise Ratio) is used in the simulations.

### 5.1. Performance of CNN-GRU

In this subsection, to test the performance in a complicated harmonic noise environment, the CNN-GRU is tested with the noise of different SNRs. To provide a fair comparison, with the same noise source and acoustic environment, average noise reduction (ANR) curves are used to evaluate the performance of the mentioned FxLMS, FLANN, DANC, and proposed CNN-GRU. Moreover, for the DANC and CNN-GRU, the target noise has three parts: the record training signal, the complicated harmonic signal, and Gaussian noise. Specifically, there is an unknown complicated harmonic noise that needs to be predicted first by the DANC and CNN-GRU-Net.

The averaged noise reduction (ANR) is used to assess the performance of different methods, and error frequency analysis is also provided. The ANR is calculated as(65)$$\mathrm{ANR}\left(n\right)=20\log\left(\frac{A_{e}\left(n\right)}{A_{d}\left(n\right)}\right),$$
where(66)$$A_{e}\left(n\right)=\gamma A_{e}\left(n-1\right)+\left(1-\gamma \right)\left|e\left(n\right)\right|,$$
and(67)$$A_{d}\left(n\right)=\gamma A_{d}\left(n-1\right)+\left(1-\gamma \right)\left|d\left(n\right)\right|.$$
and γ=0.999.

As shown in the Figure 12, the SNR is set as 25 dB. The FxLMS achieves −4.6 dB noise attenuation at 10 s. In addition, the FLANN achieves −11.3 dB noise attenuation at 10 s. As for the deep learning method DANC and CNN-GRU, the DANC achieves −36.7 dB and the CNN-GRU −31.7 dB attenuation level at 10 s. More exactly, as shown in the spectrum analysis in Figure 12, the FxLMS could not track complicated noise well. Compared with the FxLMS, the FLANN achieves a better attenuation level, especially for single-frequency noise. In the same simulation environment, although the DANC and CNN-GRU achieve a steady state at 0.13 s in the initial tracking, the DANC has a better attenuation level and smaller steady-state error. Because the CNN-GRU has fewer computing parameters than the DANC, if the CNN-GRU has more GRU layers, it could achieve better attenuation and robust performance.

When SNR decreases to 15 dB, as shown in Figure 13, the FxLMS, FLANN, DANC, and proposed CNN-GRU achieve −7.2 dB, −14.7 dB, −27.4 dB, and −31.3 dB noise attenuation at 10 s, respectively. With the decrease of SNR, the power of the training signal also decreases. Hence, the attenuation level of DANC decreased. However, for the FxLMS and FLANN, the power of complicated harmonic noise is decreasing, and the attenuation performance becomes better. The proposed CNN-GRU focuses on all the main components of the noise. Thus, the CNN-GRU still maintains a good attenuation performance. Meanwhile, the robustness of the steady state still needs to be improved because the multi-head attention filter has some noise characteristics. Moreover, as for spectrum performance in Figure 13, the FxLMS could not deal with single frequencies well. Meanwhile, the FLANN has better attenuation performance for single-frequency noise because it has multi-layer Fourier analyzers. However, as the proportion of training noise energy in the target noise decreases, the single-frequency attenuation performance of DANC and CNN-GRU decreases.

When SNR decreases to 5 dB, as shown in Figure 14, the FxLMS, FLANN, DANC, and proposed CNN-GRU achieve −9.4 dB, −17.3 dB, −11.9 dB, and −16.7 dB noise attenuation at 10 s, respectively. Because the power of complicated harmonic noise decreases, the FLANN has the best attenuation performance. In addition, compared spectrum performance in Figure 14, the FLANN still has the best single-frequency attenuation performance. However, the frequency information must be known for the FLANN; otherwise, frequency shift and frequency estimation errors will lead the FLANN algorithm to diverge. Compared the performance of FxLMS, FLANN, DANC, and CNN-GRU in Figure 12, Figure 13 and Figure 14, the proposed CNN-GRU has the best performance in when SNR >5 dB.

As shown in Figure 12, Figure 13 and Figure 14, residual harmonic noise persists after denoising using DANC and CNN-GRU, and this phenomenon becomes more significant as the signal-to-noise ratio decreases. This occurs because both DANC and CNN-GRU are pre-trained with noisy recordings and do not consider different signal-to-noise ratios. Consequently, as the signal-to-noise ratio drops, the noise frequency components added in the simulation show more randomness than the noisy recordings, which relates to the added Gaussian white noise.

### 5.2. Performance of ASL-FxLMS

To test the ASL-FxLMS (Adaptive Self-Loading FxLMS) algorithm separately, the FxLMS [27] and FxNLMS [38] are set as comparisons. To provide a fair comparison, the FxLMS, FxNLMS, and ASL-FxLMS are set with different step sizes to achieve nearly the same attenuation performance in steady state. The engine and babble noise are combined as the signal; meanwhile, 15 dB Gaussian noise is also added. As shown in Figure 15, the FxLMS, FxNLMS, and ASL-FxLMS achieve nearly −15 dB attenuation level in steady state. The FxLMS achieves the best initial tracking rate of the three methods; it reaches steady state at 2.3 s. In particular, the FxNLMS often has a better initial tracking rate than the FxLMS. However, due to the presence of strong complicated harmonic noise in the reference noise, its power fluctuates. Hence, its step size changes in a big range, which decreases the tracking rate, which leads the FxNLMS to reach a steady state at 3.2 s. While the FxLMS uses a competitive big step size, which is set by repeated experiments. That is why the FxLMS has better initial tracking than the FxNLMS in this simulation. As for the proposed ASL-FxLMS, it reaches steady state at 7.8 s. The adaptive self-loading filter decreases the tracking performance because a serial Fourier analyzer is introduced to improve its robustness in a complicated harmonic noise environment.

On the other hand, the robustness performance is very important for the ASL-FxLMS. It needs to work with reference and error interference noise. More exactly, the reference interference noise usually occurs and disappears very fast in the reference signal, which is part of the reference noise and is hard to track. In addition, error interference is usually combined with random noise and burst noise in the error signal. Especially when ASL-FxLMS works in parallel with the CNN-GRU-Net, the periodical component may be almost attenuated by the CNN-GRU. Then, the Gaussian noise will become fatal to the robustness of the traditional adaptive filter. Hence, the FxLMS, FxNLMS, and the proposed ASL-FxLMS need to be tested in the acoustic environment with reference and error interference to verify their robust performance.

The reference interference, which includes whistling and Gaussian noise, is introduced from 8.6 s to 10.5 s as shown in Figure 16. The FxLMS diverges at 8.6 s because it cannot deal with impulsive interference. The FxNLMS achieves robust tracking when the reference interference occurs and when it disappears. As for the ASL-FxLMS, it remains robust from 8.6 s to 12 s.

Moreover, as shown in Figure 17, the error interference will also decrease the performance of ANC algorithms. Thus, the whistling and Gaussian noise interference are added to the error signal from 8.6 s to 10.5 s. Meanwhile, the FxLMS diverges at 8.6 s. Although the FxNLMS achieves robust tracking, it fails to reach the best attenuation level from 10.5 s to 12 s. In comparison, the proposed ASL-FxLMS still achieves robust tracking even when the interference is added, which gives further proof that the ASL-FxLMS has better robust performance than the FxLMS and FxNLMS in the simulations.

### 5.3. Performance of Hybrid ASL-FxLMS and CNN-GRU

To balance the real-time, robustness, and attenuation performance, the CNN-GRU and ASL-FxLMS are combined in parallel. Moreover, the hybrid of CNN-GRU and ASL-FxLMS net is briefly renamed as the Hybrid net in simulations in this subsection. The reference and error interference are introduced to test the performance of the Hybrid net. Furthermore, to provide a fair comparison, the parameters of ASL-FxLMS, CNN-GRU, and the Hybrid net are selected by multiple experiments.

As shown in Figure 18, the reference interference is introduced from 8.6 s to 10 s as the same simulation results in Section 5.2. The ASL-FxLMS needs two adaptive self-loading sine periods to recover the normal tracking rate. The CNN-GRU and the Hybrid net perform similarly in steady-state attenuation level, achieving −26.3 dB and −27.2 dB, respectively. However, compared with the Hybrid net, the CNN-GRU has a bigger misleading peak during the occurrence of reference interference. More exactly, the average ANR of CNN-GRU is 3.0 dB from 8.6 s to 10 s, while the average ANR of the Hybrid net is −21.4 dB from 8.6 s to 10 s. Actually, this advantage benefits from the convex combination function $\lambda \left(n\right)$ as shown in Equation (Equation 13), which can effectively integrate the advantages of ASL-FxLMS and CNN-GRU.

When the error interference is introduced as shown in Figure 19, during the occurrence of error interference, the ASL-FxLMS, CNN-GRU, and Hybrid net have 10.6 dB, 3.4 dB, and −20.3 dB average misleading peak. Moreover, the Hybrid net has the best −27.4 dB attenuation level, followed by CNN-GRU with −25.8 dB. The mentioned also proves the Hybrid net has a balanced performance of competitive good tracking, high attenuation level, and robust steady state.

### 5.4. Time-Series and Frequency ANC Analysis for Various Types of Noise

This section evaluates the proposed algorithm’s noise reduction performance across different noise types. Specifically, speech noise, truck noise, and building noise are used as examples to assess its capabilities. Additionally, we analyze the average noise reduction results for these three noise categories.

#### 5.4.1. Adaptability of the Proposed Algorithm to Various Types of Noise

To verify the adaptability of the proposed algorithm to various noises, noise clips of speech, truck, and building noise were selected. The analysis in both the time and frequency domains before and after ANC noise reduction was displayed. As shown in Figure 20, Figure 21 and Figure 22, speech, truck, and building noise are used to test the attenuation capability of the proposed method. The original time-series and spectrum are shown with a blue line, and the time-series and spectrum after the ANC process are represented by a red line.

In Figure 20a, the maximum normalized amplitude is 0.52, while the maximum normalized amplitude after the ANC process is 0.18. As shown in Figure 20b, the noise within 1500 Hz has an average noise reduction of 22.83 dB relative to the initial speech noise. However, the noise reduction effect near 420 Hz on the spectrum is poor, and the noise level increases by 6.73 dB compared to the original speech noise.

In Figure 21a, the maximum normalized amplitude is 0.41, while the maximum normalized amplitude after the ANC process drops to 0.19. As shown in Figure 21b, the noise within 1500 Hz is reduced by an average of 14.59 dB compared to the truck noise. However, the noise reduction near 870 Hz on the spectrum is ineffective, and the noise level increases by 13.22 dB compared to the original truck noise.

In Figure 22a, the maximum normalized amplitude is 0.47, while after the ANC process, it decreases to 0.21. As shown in Figure 22b, the noise below 1500 Hz has an average noise reduction of 7.33 dB compared to the initial building noise. Meanwhile, the power around 710 Hz in the spectrum is 13.22 dB higher than the original building noise.

The above simulation results show that although the noise reduction ability of the proposed method varies with different noises, the effective noise reduction capability of the algorithm has been confirmed in both the time domain and the frequency domain.

#### 5.4.2. Average Noise Reduction Analysis

To further verify the algorithm’s noise reduction ability for different noises and compare multiple types of ANC algorithms under the same conditions, the simulations shown in Figure 23, Figure 24 and Figure 25 are added to demonstrate the advantages of the proposed method. To ensure a fair comparison, all ANC algorithms are adjusted to their optimal parameters under non-misadjustment conditions to achieve the maximum noise reduction level.

As shown in Figure 23, speech noise is used as the target noise, and FxLMS [39], FxRLS [40], FxRMC [41], FxGMN [38], GFANC-Kalman [42], DANC [15], and the proposed method are employed as ANC noise reduction algorithms. The FxLMS reaches steady state at 3.3 s and achieves a maximum attenuation level of 5.2 dB. The FxRLS reaches steady state at 2.7 s and achieves a maximum attenuation level of 10.6 dB. The FxRMC reaches steady state at 3.5 s and achieves a maximum attenuation level of 13.7 dB. The FxGMN reaches steady state at 8 s and achieves a maximum attenuation level of 10.5 dB. The GFANC-Kalman reaches steady state at 1.4 s and achieves a maximum attenuation level of 16.1 dB. The DANC reaches steady state at 0.17 s and achieves a maximum attenuation level of 22.9 dB. The proposed method reaches steady state at 0.17 s and achieves a maximum attenuation level of 23.3 dB.

As shown in Figure 24, for the truck noise, the FxLMS reaches steady state at 2.2 s and attains a maximum attenuation of 3.1 dB. The FxRLS reaches steady state at 3.5 s with a maximum attenuation of 8.2 dB. The FxRMC reaches steady state at 3.8 s and achieves 12.4 dB maximum attenuation. The FxGMN reaches steady state at 7 s with a maximum attenuation of 8.1 dB. The GFANC-Kalman reaches steady state at 0.8 s with a maximum attenuation of 12.1 dB. The DANC reaches steady state at 0.9 s and achieves 18.1 dB maximum attenuation. However, its robust performance still needs improvement. The proposed method reaches steady state at 0.7 s with a maximum attenuation of 18.7 dB.

As shown in Figure 25, for building noise, the FxLMS reaches steady state at 2.1 s and achieves a maximum attenuation of 1.3 dB. The FxRLS reaches steady state at 4 s with a maximum attenuation of 4.2 dB. The FxRMC reaches steady state at 4 s with a maximum attenuation of 8.7 dB. The FxGMN reaches steady state at 9 s and achieves a maximum attenuation of 3.2 dB. The GFANC-Kalman reaches steady state at 4 s with a maximum attenuation of 8.8 dB. The DANC reaches steady state at 0.13 s with a maximum attenuation of 10.1 dB. The proposed method reaches steady state at 0.13 s and achieves a maximum attenuation of 13.3 dB. Under the same conditions, the time costs of each mentioned algorithm are listed in Table 5.

In summary, the simulation comparison results show that the FxLMS algorithm [39] has the lowest computational load, but its noise reduction effect on various types of noise is not very effective. Compared with the FxLMS algorithm, the RLS algorithm [40,41] adds a Gaussian kernel function that offers better stability and increases the computational load to improve noise reduction. However, it is necessary to test multiple sets of different parameters to adapt to various noise environments and ensure system stability. The hybrid structure algorithm [38] combines the advantages of the two substructures, but selecting the optimal parameters when dealing with different noise environments remains challenging. The deep network algorithm [15,42] generally achieves higher noise reduction through pre-training. By improving the network structure and integrating methods such as Kalman adaptive filters, noise reduction performance and stability can be significantly enhanced. The simulation results indicate that the proposed algorithm has a computational load similar to that of the deep network method but offers better stability and noise reduction performance. It also demonstrates the superiority of the hybrid filter structure in the ANC system. However, the deep network algorithm still needs to reduce computational load further to meet practical application requirements.

## 6. Conclusions

Based on the bending-pipe model of an air conditioner, a hybrid CNN-GRU and adaptive self-loading FxLMS (ASL-FxLMS) structure is proposed to achieve more efficient complex harmonic noise attenuation with an ANC system. The CNN-GRU network provides a high attenuation level, especially for various types of noise that can be trained. While the CNN-GRU network attains significant noise reduction, it may leave residual harmonic noise when the signal-to-noise ratio is low. Simultaneously, the parallel setup of ASL-FxLMS can robustly track uncertain, complex harmonic noise. However, the noise reduction performance of the ASL-FxLMS is limited for signals with many single-frequency and uncertain harmonic frequency components. Moreover, the time-domain and frequency-domain simulation analysis results have demonstrated that the proposed hybrid network balances tracking, noise reduction, and stability performance. Furthermore, in future practical applications, such as when air conditioning noise mixes with mechanical noise, building noise, and outdoor speech noise, the proposed hybrid ASL-FxLMS and CNN-GRU network could deliver even better noise reduction performance.

## Figures and Tables

**Figure 1 sensors-25-06293-f001:**
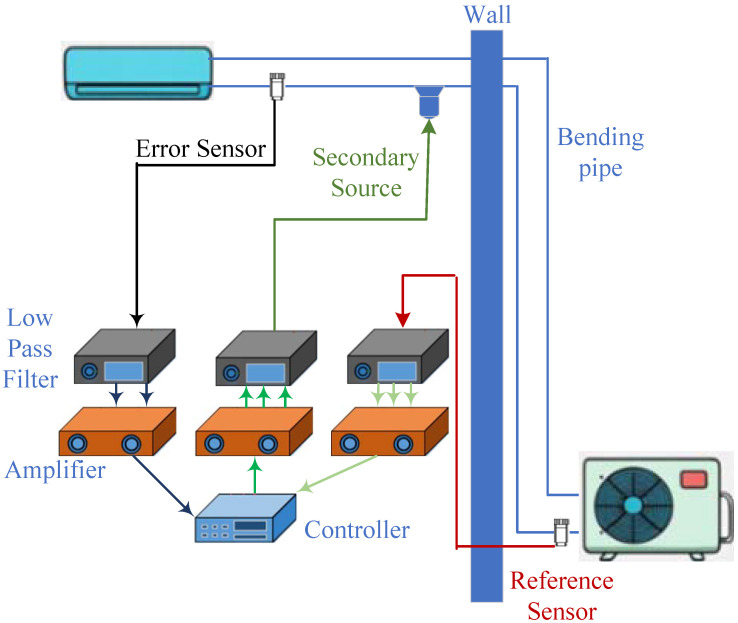
Brief diagram of the air-conditioner ANC system.

**Figure 2 sensors-25-06293-f002:**
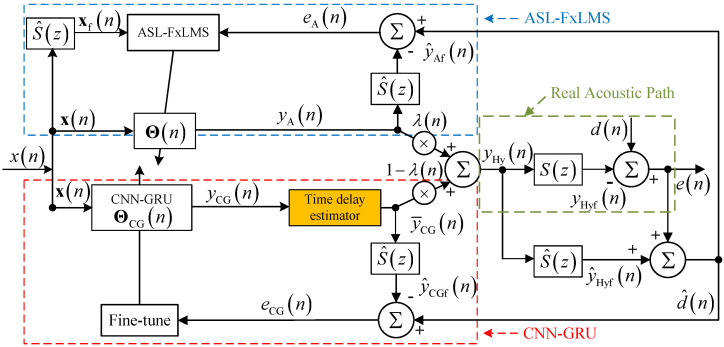
Brief diagram of a hybrid adaptive filter and deep network.

**Figure 3 sensors-25-06293-f003:**
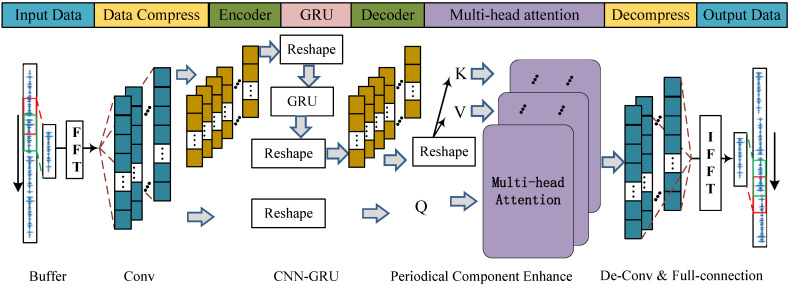
Structure of the CNN-GRU.

**Figure 4 sensors-25-06293-f004:**
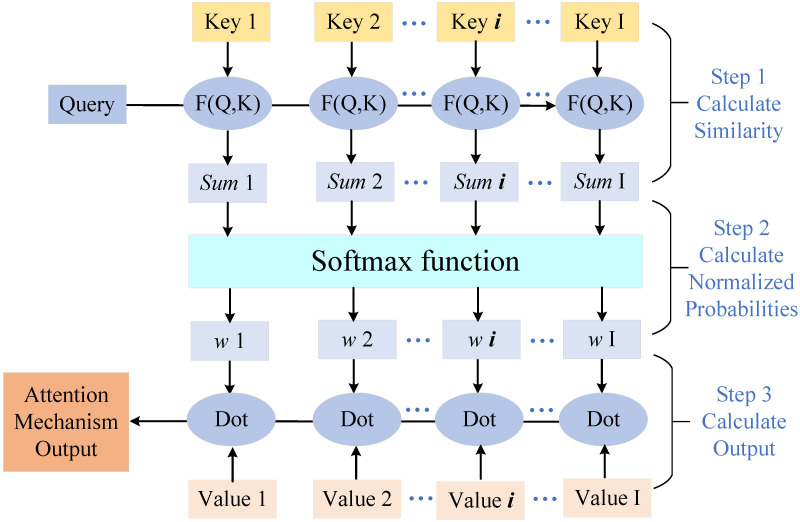
Process of multi-head attention layer.

**Figure 5 sensors-25-06293-f005:**
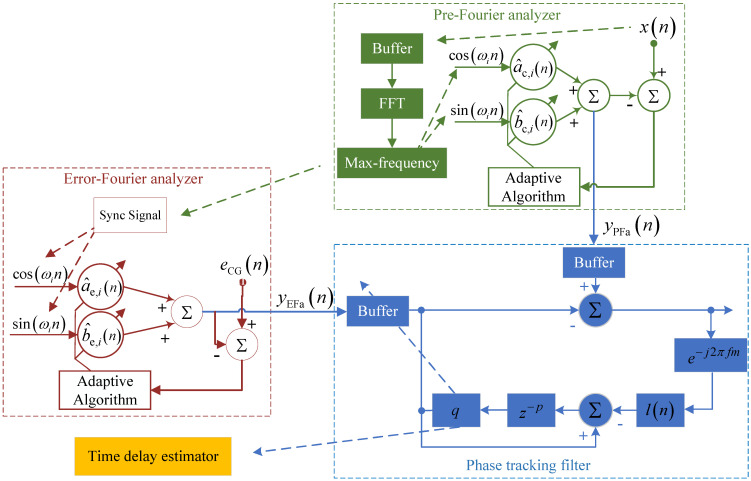
Brief diagram of phase tracking process.

**Figure 6 sensors-25-06293-f006:**
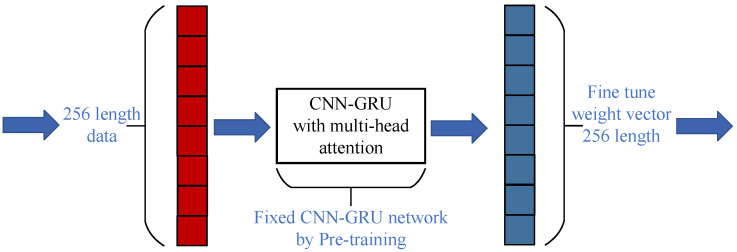
Connections of CNN-GRU and fine-tune.

**Figure 7 sensors-25-06293-f007:**
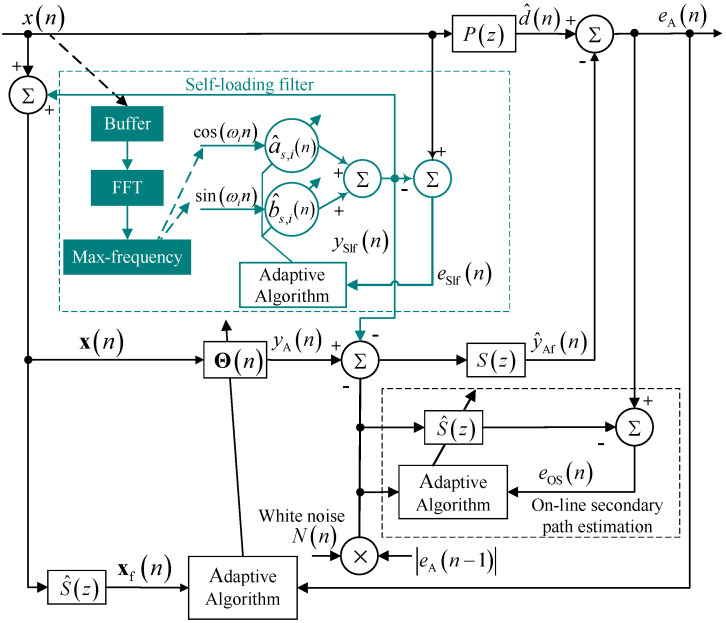
Brief diagram of adaptive self-loading FxLMS (ASL-FxLMS).

**Figure 8 sensors-25-06293-f008:**
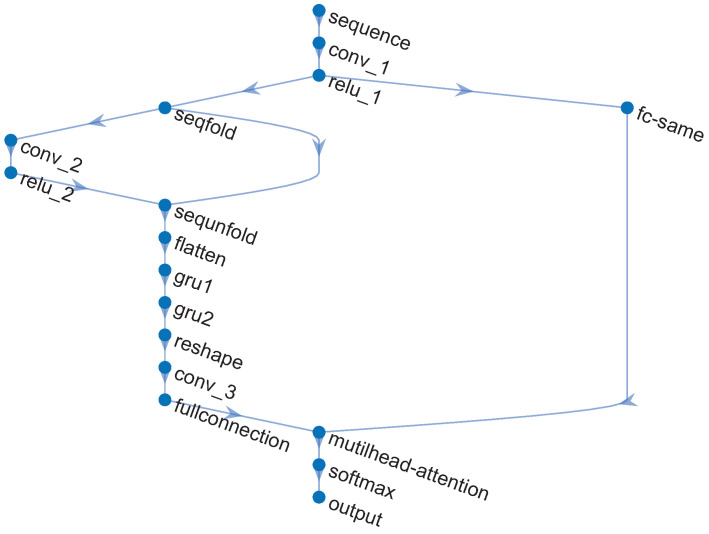
Layers connection of CNN-GRU with multi-head-attention.

**Figure 9 sensors-25-06293-f009:**
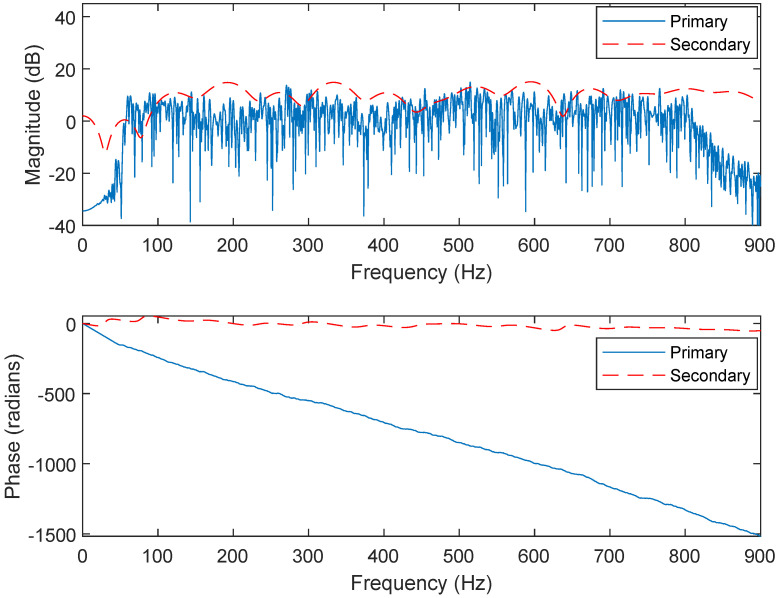
Frequency responses of the primary and secondary paths before changing.

**Figure 10 sensors-25-06293-f010:**
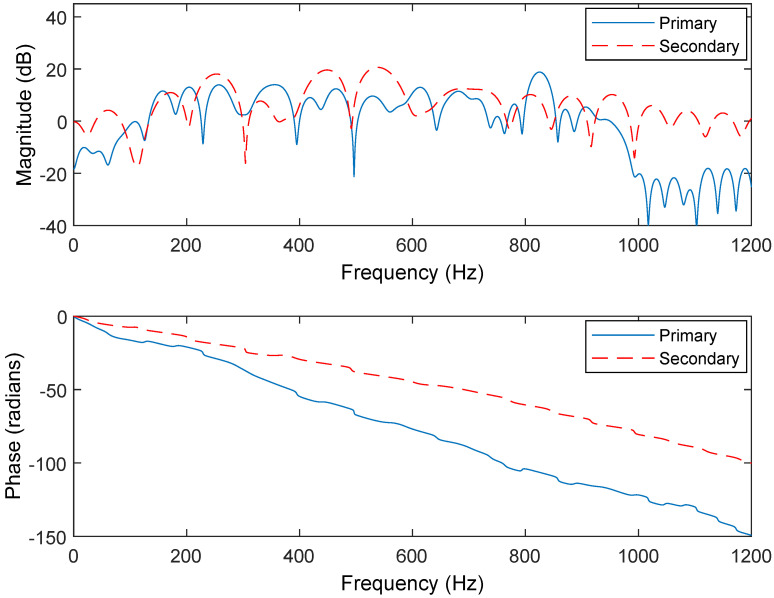
Frequency responses of the primary and secondary paths after changing.

**Figure 11 sensors-25-06293-f011:**
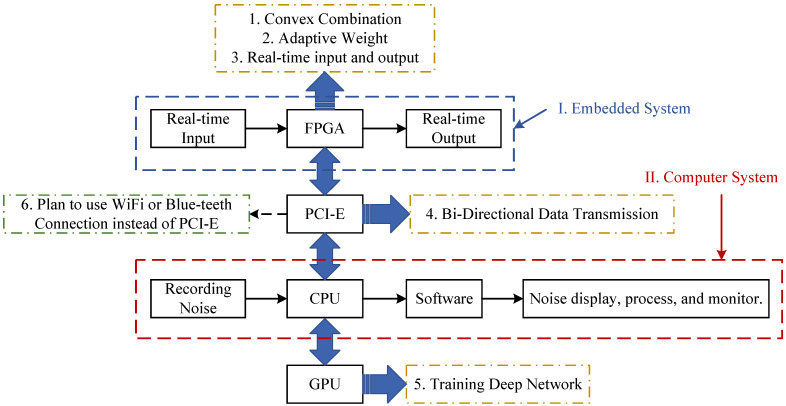
Block diagram of hybrid embedded and computer ANC system.

**Figure 12 sensors-25-06293-f012:**
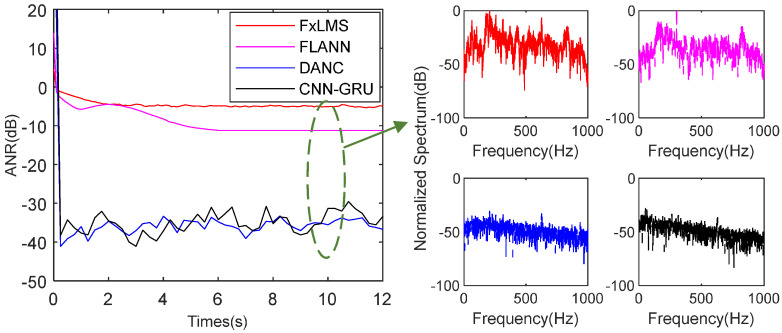
ANR and spectrum performance of the mentioned methods with SNR 25 dB.

**Figure 13 sensors-25-06293-f013:**
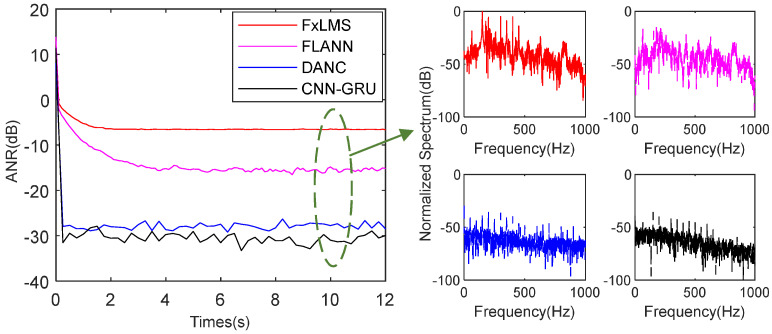
ANR and spectrum performance of the mentioned methods with SNR 15 dB.

**Figure 14 sensors-25-06293-f014:**
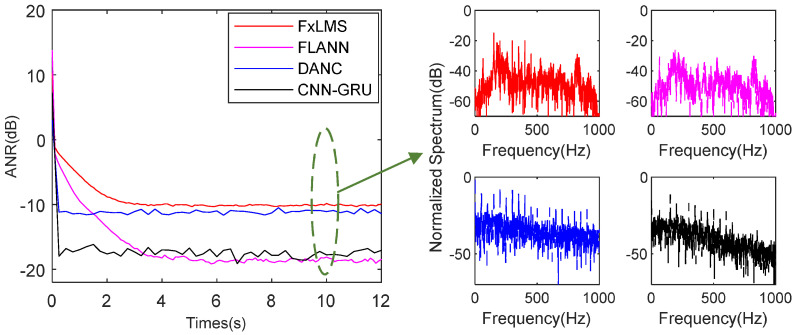
ANR and spectrum performance of the mentioned methods with SNR 5 dB.

**Figure 15 sensors-25-06293-f015:**
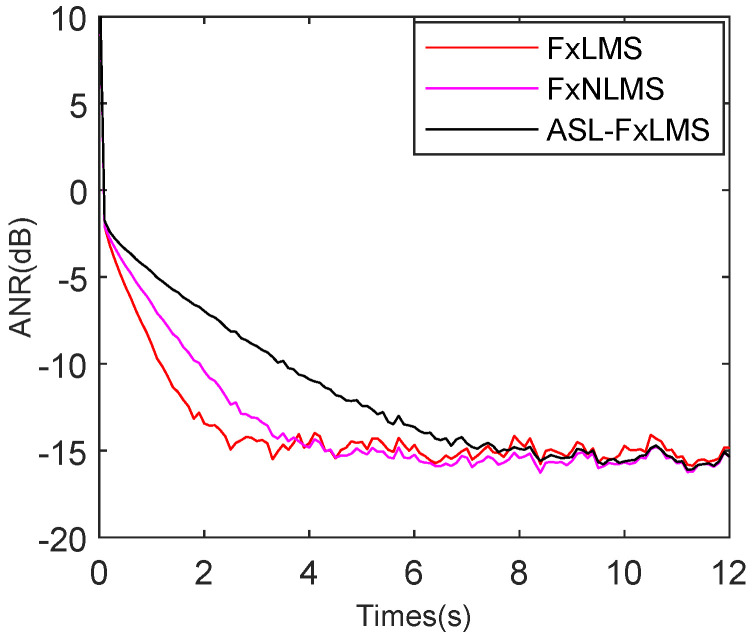
ANR performance comparison for FxLMS, FxNLMS, and ASL-FxLMS.

**Figure 16 sensors-25-06293-f016:**
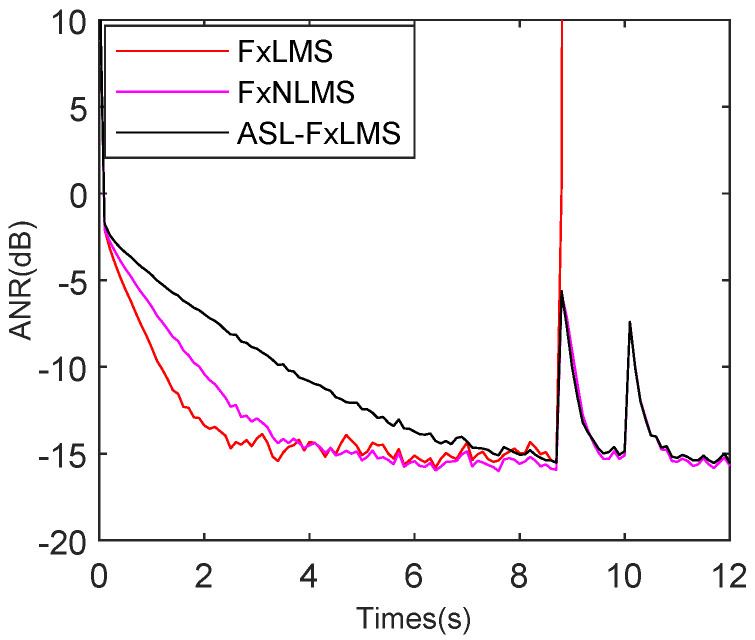
ANR performance comparison with reference interference for FxLMS, FxNLMS, and ASL-FxLMS.

**Figure 17 sensors-25-06293-f017:**
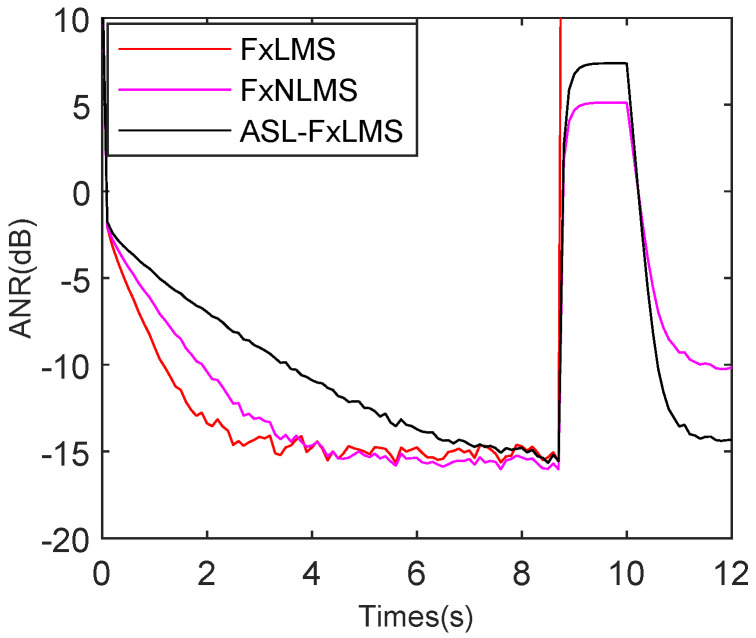
ANR performance comparison with error interference for FxLMS, FxNLMS, and ASL-FxLMS.

**Figure 18 sensors-25-06293-f018:**
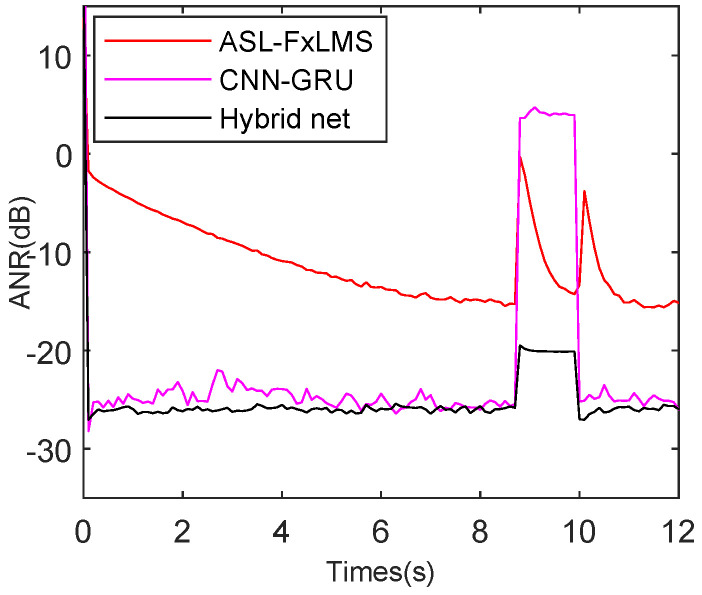
ANR performance comparison with reference interference for ASL-FxLMS, CNN-GRU, and hybrid net of ASL-FxLMS-CNN-GRU.

**Figure 19 sensors-25-06293-f019:**
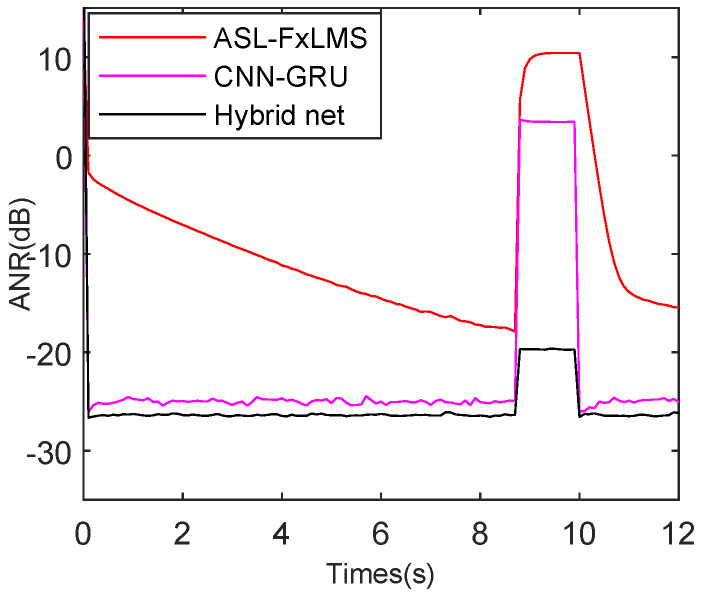
ANR performance comparison with error interference for ASL-FxLMS, CNN-GRU, and hybrid net of ASL-FxLMS-CNN-GRU.

**Figure 20 sensors-25-06293-f020:**
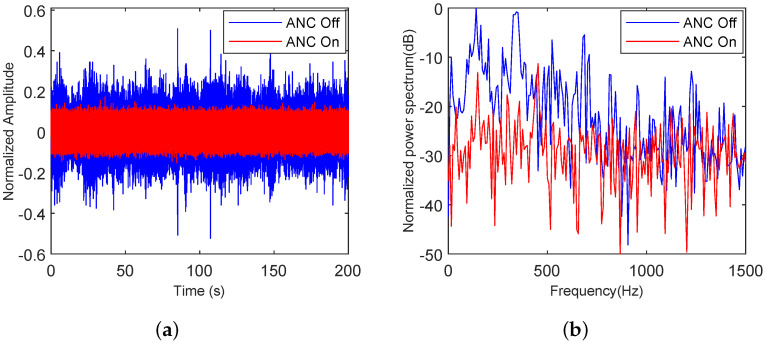
Speech noise analysis with the proposed ANC method turned off and on, (**a**) in the time domain, (**b**) in the frequency domain.

**Figure 21 sensors-25-06293-f021:**
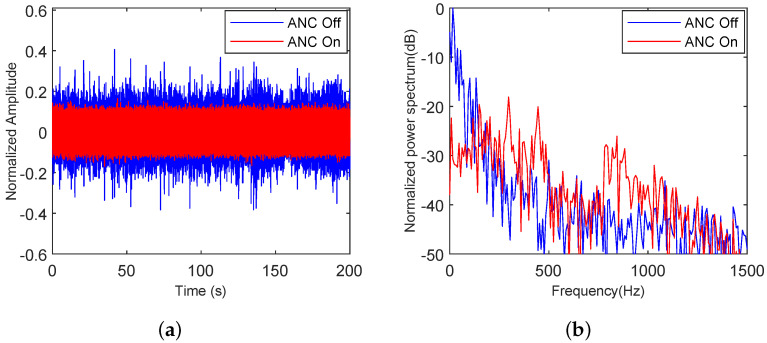
Truck noise analysis with the proposed ANC method turned off and on, (**a**) in the time domain, (**b**) in the frequency domain.

**Figure 22 sensors-25-06293-f022:**
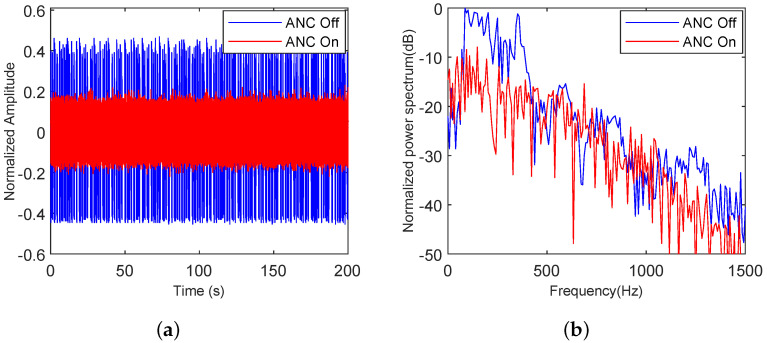
Building noise analysis with the proposed ANC method turned off and on, (**a**) in the time domain, (**b**) in the frequency domain.

**Figure 23 sensors-25-06293-f023:**
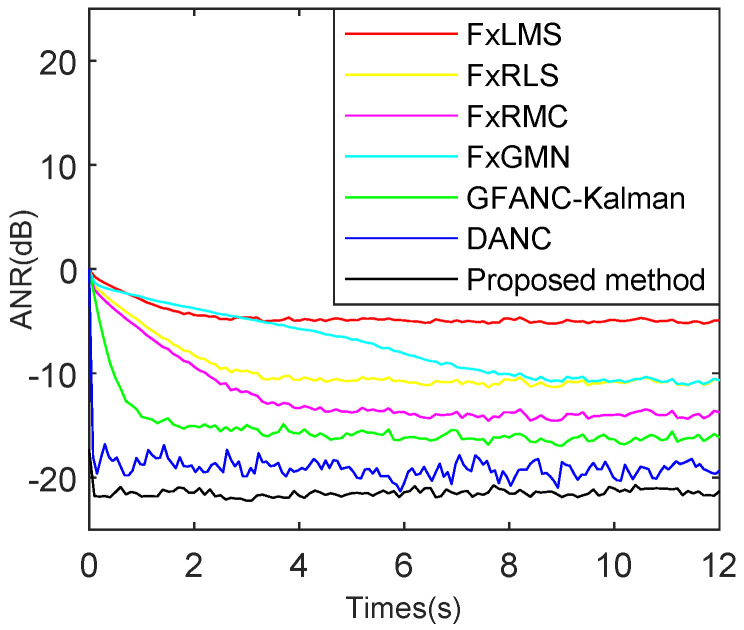
Comparison of ANR (Average Noise Reduction) curves for various ANC algorithms, for combined speech and Gaussian noise with an SNR (Signal-to-Noise Ratio) of 15 dB.

**Figure 24 sensors-25-06293-f024:**
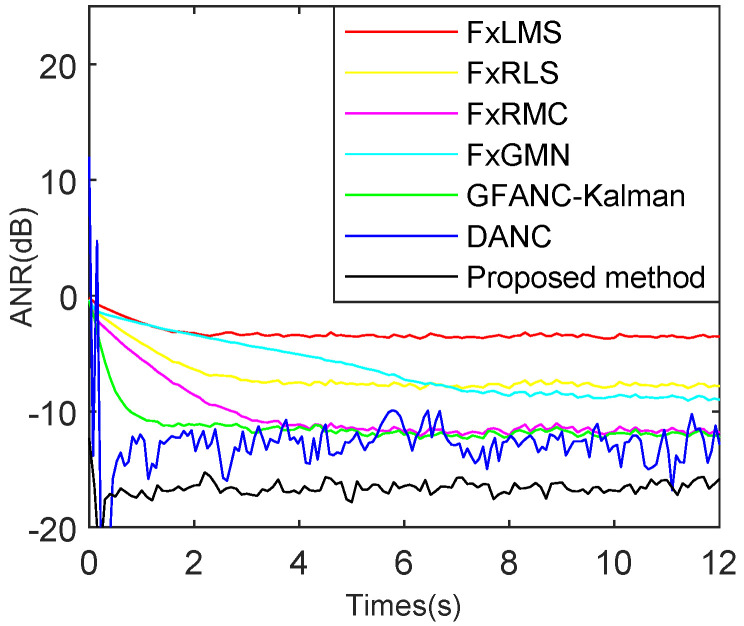
Comparison of ANR curves for various ANC algorithms, for combined truck and Gaussian noise with an SNR of 15 dB.

**Figure 25 sensors-25-06293-f025:**
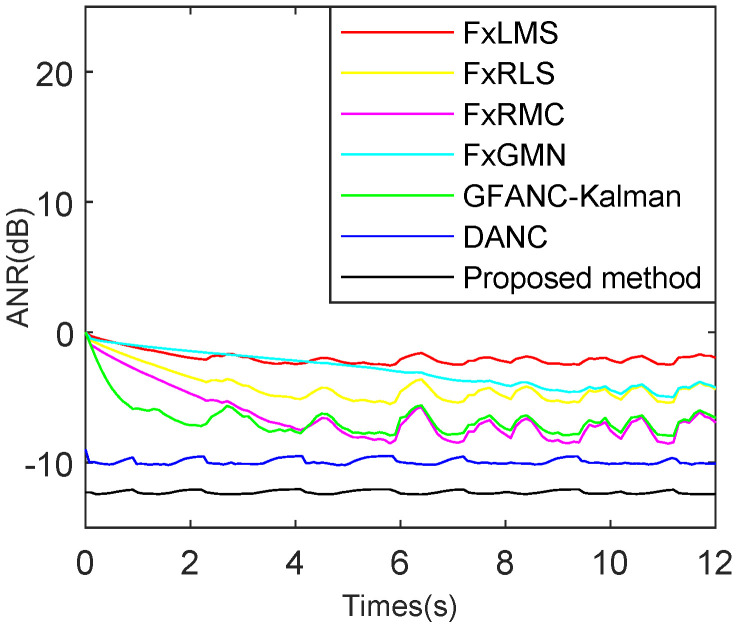
Comparison of ANR curves for various ANC algorithms, for combined building and Gaussian noise with an SNR of 15 dB.

**Table 1 sensors-25-06293-t001:** Notation and Definitions.

Notation	Definitions
$d\left(n\right)$	Reference signal at error sensor
$\hat{d}\left(n\right)$	Estimated reference signal at error sensor
$\lambda \left(n\right)$	Output coefficient of convex combination
$e\left(n\right)$	Real error signal
${\hat{y}}_{\mathrm{Hyf}}\left(n\right)$	Hybrid net output filtered by estimated secondary path
$x\left(n\right)$	Reference input signal vector
$e_{\mathrm{CG}}\left(n\right)$	Error signal of CNN-GRU
${\hat{y}}_{\mathrm{CGf}}\left(n\right)$	CNN-GRU Output filtered by estimated secondary path
$x_{f}\left(n\right)$	Reference signal filtered by estimated secondary path
$\Theta \left(n\right)$	Weight vector of ASL-FxLMS
${\hat{a}}_{s,j}\left(n\right)$	Weight vector of adaptive self-loading filter of ASL-FxLMS
${\hat{b}}_{s,j}\left(n\right)$	Weight vector of adaptive self-loading filter of ASL-FxLMS
$e_{A}\left(n\right)$	Error signal of ASL-FxLMS

**Table 2 sensors-25-06293-t002:** Input size, output size, and hyperparameters of each layer.

Layer Name	Input Size	Hyperparameters	Output Size
Sequence	1×T×256	—	256×T×1
Conv1	256×T×1	3×1, $\left(1,4\right)$, 64	64×T×4
Relu1	64×T×4	—	64×T×4
Seqfold	64×T×4	Mini-Batch-Size	64×T×4
Conv2	64×T×4	3×1, $\left(1,4\right)$, 16	16×T×16
Relu2	16×T×16	—	16×T×16
SeqUnfold	16×T×16	Mini-Batch-Size	16×T×16
Flatten	1×T×256	—	1×T×256
GRU1	1×T×256	256	1×T×256
GRU2	1×T×256	256	1×T×256
Reshape	1×T×256	—	16×T×16
Conv3	16×T×16	3×1, $\left(1,4\right)$, 64	64×T×4
Full-Connection	64×T×4	—	256×T×1
Multi-head Attention	256×T×1	—	256×T×1
Softmax	256×T×1	—	256×T×1
Output	256×T×1	—	1×T×256

**Table 3 sensors-25-06293-t003:** The computational cost of each mentioned algorithm in one step.

Name	Addition	Multiplication	Exponent
FxLMS	512	641	No
FxNLMS	768	899	No
ASL-FxLMS	516	645	No
FLANN	6782	7385	No
DANC	21,249	10,977	No
CNN-GRU	15,328	6836	3

**Table 4 sensors-25-06293-t004:** Main operation of proposed hybrid CNN-GRU and ASL-FxLMS.

**Parameters and Preparation.**
Train CNN-GRU $\Theta_{\mathrm{CG}}\left(n\right)$ with noise of multi-sources.
Step size $\mu_{A}$, $\mu_{\mathrm{PFa}}$, $\mu_{\mathrm{EFa}}$, $\mu_{\mathrm{Slf}}$, $\mu_{\sigma }$, and $\mu_{\mathrm{OS}}$.
Initiate the ANC system by estimating the estimated secondary path $\hat{S}\left(z\right)$
and max-power frequency of the adaptive self-loading structure.
**Main Operations.**
**(ASL-FxLMS)**
1. Reference signal:
$x\left(n\right)=\left[x\left(n\right)+y_{\mathrm{Slf}}\left(n\right),x\left(n-1\right)+y_{\mathrm{Slf}}\left(n-1\right),\ldots ,x\left(n-m_{x}+1\right)+y_{\mathrm{Slf}}\left(n-m_{x}+1\right)\right]$
2. ASL-FxLMS output: $y_{A}\left(n\right)=\Theta^{T}\left(n\right)x\left(n\right)$
3. Error signal: $e_{A}\left(n\right)=\hat{d}\left(n\right)-y_{\mathrm{AF}}\left(n\right)$
4. Update weight vector: $\Theta \left(n+1\right)=\Theta \left(n\right)+\mu_{A}e_{A}\left(n\right)x_{f}\left(n\right)$
5. Update adaptive self-loading filters: ${\hat{a}}_{s,i}\left(n+1\right)={\hat{a}}_{s,i}\left(n\right)+\mu_{\mathrm{Slf}}x_{ai}\left(n\right)e_{\mathrm{Slf}}\left(n\right)$
${\hat{b}}_{s,i}\left(n+1\right)={\hat{b}}_{s,i}\left(n\right)+\mu_{\mathrm{Slf}}x_{bi}\left(n\right)e_{\mathrm{Slf}}\left(n\right)$
6. Output of adaptive self-loading filters: $y_{\mathrm{slf}}\left(n\right)={\hat{a}}_{s,j}\left(n\right)x_{ai}\left(n\right)+{\hat{b}}_{s,j}\left(n\right)x_{bi}\left(n\right)$
**(CNN-GRU)**
7. Data input: Put $\left[x\left(n\right),x\left(n-1\right),\ldots ,x\left(n-T+1\right)\right]$ into CNN-GRU-Net.
8. Estimate time-delay: Put $y_{\mathrm{CG}}\left(n\right)$ into Time-delay estimator and obtain ${\bar{y}}_{\mathrm{CG}}\left(n\right)$.
9. Get error signal for fine-tune: $e_{\mathrm{CG}}\left(n\right)=\hat{d}\left(n\right)-{\hat{y}}_{\mathrm{CGf}}\left(n\right)$
**(Convex Combination)**
9. Calculate output: $y_{\mathrm{Hy}}\left(n\right)=\lambda \left(n\right)y_{A}\left(n\right)+\left[1-\lambda \left(n\right)\right]{\bar{y}}_{\mathrm{CG}}\left(n\right)$
10. Update partition coefficient: $\lambda \left(n\right)=\frac{1}{1+e^{-\sigma \left(n\right)}}$
11. Estimate power: $\sigma \left(n+1\right)=\sigma \left(n\right)+\mu_{\sigma }\frac{2e\left(n\right)}{{\left[1+e^{2}\left(n\right)\right]}^{2}}\left[{\bar{y}}_{\mathrm{CG}}\left(n\right)-y_{A}\left(n\right)\right]\lambda \left(n\right)\left[1-\lambda \left(n\right)\right]$

**Table 5 sensors-25-06293-t005:** Time cost of training and tracking with MATLAB 2025, NVIDIA 4060 (Santa Clara, CA, USA), and Intel I7 core CPU (Santa Clara, CA, USA).

	Training Cost (Hours)	Cost of ANC Process for 200 s Noise (Seconds)
FxLMS	0	312
FxRLS	0	940
FxRMC	0	1711
FxGMN	0	2691
GFANC-Kalman	14	2067
DANC	22	3987
Propose method	16	2670

## Data Availability

Data are contained within the article.

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
