# Peer review of "A New Hybrid Adaptive Self-Loading Filter and GRU-Net for Active Noise Control in a Right-Angle Bending Pipe of an Air Conditioner"

_sensors, 2025, doi:10.3390/s25206293_

Round 1
Reviewer 1 Report
Comments and Suggestions for Authors
The noise reduction of air conditioners in enclosed spaces is a significant concern. This manuscript introduces a hybrid adaptive self-loading filtered-x least mean square (ASL-FxLMS) and convolutional neural network-gate recurrent unit (CNN-GRU) network to tackle the harmonic noise problem. To enhance attenuation performance, several adaptive filters are used to enable real-time operation. Here are some suggested revisions to the manuscript.
- (Chapter Introduction, Paragraph 6) It is suggested that the author highlight the advantages of employing the convex combination structure in the proposed method. Why is this structure necessary?
- (Chapter 2.1) The author should provide a clearer illustration of how to utilize the bending pipe model.
- (Table 1) The format of Table 1 requires adjustment, as the current width is insufficient.
- (Chapter 3.1, Figure 3) Additional descriptions should be included for Figure 3 to clarify the architectural details of the CNN-GRU model.
- (Figures 13, 14, and 15) More detailed frequency analysis needs to be incorporated. For instance, why does significant harmonic noise still persist? Additionally, the sizes of these figures need correction.
- (Conclusion) The author should clearly summarize the results of the proposed hybrid structure in the conclusion, including its advantages and limitations compared to traditional methods, and discuss potential future applications and developments of the technology.
Author Response
Please find the detailed changes in the PDF file.

Reviewer 2 Report
Comments and Suggestions for Authors
The submitted draft reports ANC with a hybrid CNN-GRU and adaptive self-loading FxLMS algorithm. The application area is on the air conditioner bends. It is suggested that the quiet space is especially required for patients during recovery. The proposed algorithm may have applications in such area. For publication it is recommended the following major revision is done.
- The verification of the proposed algorithm was performed using the simulation model. The ANC performance is strongly dependent on the noise spectral and transient characteristics. It is recommended to show some noise samples and use the measured sample for the verification of the performance.
- The simulation shows the performance only in the level. It is crucial to show the performance with respect to computing power. Some sample is recommended to be included how long it takes to show the performance.
Author Response
Reviewer 2:
(Please find the detailed changes in the PDF file.)
Comments and Suggestions for Authors:The submitted draft reports ANC with a hybrid CNN-GRU and adaptive self-loading FxLMS algorithm. The application area is on the air conditioner bends. It is suggested that the quiet space is especially required for patients during recovery. The proposed algorithm may have applications in such area. For publication it is recommended the following major revision is done.
- The verification of the proposed algorithm was performed using the simulation model. The ANC performance is strongly dependent on the noise spectral and transient characteristics. It is recommended to show some noise samples and use the measured sample for the verification of the performance.
Reply:
Thanks for the suggestion. It is very important to verify and clearly demonstrate the noise reduction performance of the proposed algorithm. Although the proposed work compares different algorithms and shows the simulation details of spectrum noise reduction, it is essential to include tests with different types of noise to validate the algorithm's effectiveness. Therefore, this paper will add a simulation analysis of the proposed algorithm's adaptability to various noise types. The simulation will display noise samples in both time and frequency domains, along with the measured sound samples after the ANC process.
- The simulation shows the performance only in the level. It is crucial to show the performance with respect to computing power. Some sample is recommended to be included how long it takes to show the performance.
Reply:
Thanks for the suggestion. The time cost of training and tracking is indeed very important for the ANC system. Therefore, a time cost table is included, as shown in Table 1.

Reviewer 3 Report
Comments and Suggestions for Authors
This manuscript presents a hybrid ANC approach and validates the conclusion with a set of simulation results. The design of CNN-GRU is reasonable. Overall, it is a comprehensive work that considers several aspects in ANC design. There are some major concerns:
- The real acoustic path used in this paper is linear, based on Figure 2, and equation (7). Although it was emphasized that the method is designed for non-linear cases, the setup of this acoustic system is linear.
- The usage of "complicated" in describing the harmonic noises is vague. What type of "complication"? The "harmonic noises" itself is not considered complicated in the ANC field.
- The method to be compared is not a reasonable set. One important reason FxLMS is so popular in ANC field is because of its low computational complexity. If you are using a NN inside the ANC system, you should compare the proposed method with other methods having similar computational load (or at least higher than a simple FxLMS). For example, the RLS-based method.
- Based on the proposed method, the CNN-GRU essentially serves as a preconditioning filter for ANC systems, especially considering it was trained beforehand and fixed during run time. Some comments/literature review/comparison should be done with related literature.
Author Response

(The authors gave the same response as above.)

Round 2
Reviewer 2 Report
Comments and Suggestions for Authors
The raised comments were answered successfully.
Reviewer 3 Report
Comments and Suggestions for Authors
The author addressed my questions in the revised manuscript. The manuscript is improved much after the revision. I don't have further questions on this manuscript.